# Veiled to Express: Uncovering the Effect of Mask-Wearing on Voice Behavior in the Workplace

**DOI:** 10.3390/bs14040309

**Published:** 2024-04-11

**Authors:** Ziyan Cui, Yangjie Liu, Xiaoxiao Sun, Zhe Shang, Minya Xu

**Affiliations:** 1School of Economics and Management, Tsinghua University, Beijing 100084, China; cuizy21@mails.tsinghua.edu.cn; 2School of Government, Beijing Normal University, Beijing 100875, China; lyj202312@163.com (Y.L.); 202211260006@mail.bnu.edu.cn (X.S.); 3Guanghua School of Management, Peking University, Beijing 100871, China; minyaxu@gsm.pku.edu.cn

**Keywords:** mask, voice, psychological safety

## Abstract

Since the outbreak of COVID-19, mask-wearing has become a widespread phenomenon. Even after the pandemic, people continue to maintain the habit of wearing masks in their daily lives. While existing research has explored how mask-wearing can influence wearers’ behavior in everyday life, its effects in the workplace have received less attention. Drawing on self-perception theory, this study examined the positive effect of mask-wearing in the workplace on wearers’ voice behavior via psychological safety. An online experiment (N = 291) using a within-subject manipulation of wearing masks supported our hypotheses. This study uncovered the positive psychological and behavioral consequences of mask-wearing beyond its benefits in people’s health conditions and everyday life.

## 1. Introduction

The COVID-19 pandemic, characterized by the rapid spread of the virus and the resulting urgent and widespread responses, has fundamentally reshaped public behaviors worldwide. Practices such as mask-wearing, frequent hand sanitization, and social distancing became fundamental strategies to mitigate the transmission of virus. These measures, while health-oriented, had far-reaching implications beyond their immediate health objectives, profoundly altering social interactions and psychological well-being. Particularly, as the pandemic evolved, the act of wearing a mask has transcended its initial medical purpose to become a normal daily practice with additional meanings and implications. This transformation was not merely a response to public health policies but also reflected a significant behavioral adaptation among individuals. Even after the pandemic, mask-wearing remains a prevalent and unique social common practice without mandatory requirements. In addition to serving as a physical protective barrier against the virus [1], mask can generate a range of psychological spillover effects, influencing social interactions and individual behaviors in profound ways. Existing research has examined the psychological effects of mask-wearing on wearers. For instance, masks can help people reduce the fear of negative evaluation and social anxiety in face-to-face interactions because of the self-concealing functions [2,3,4]. In addition, masks have emerged as a moral symbol that reduces wearers’ deviant behavior by heightening their moral awareness [5].

However, the existing literature has predominantly focused on the psychological effects of mask-wearing in everyday life, leaving a gap in our understanding of its implications within organizational contexts, particularly concerning how such a simple act can shape complex dynamics in the workplace. Given the psychological benefits associated with masks as mentioned above, notably the elevation of moral awareness and reduction of social anxiety, we propose that mask-wearing might foster some beneficial yet potentially risky behaviors in the workplace. To help address this knowledge gap, this study aims to examine the effect of employees’ mask-wearing in the workplace on their voice behavior, which is defined as an informal and discretionary expression of constructive opinions, concerns, or ideas about work-related issues [6,7,8,9]. On the one hand, employees’ voices have long been recognized as key drivers of positive outcomes such as team performance and crisis prevention [10,11]. On the other hand, voice behavior can be risky since it might challenge the status quo and bring forth undesirable social consequences and deplete personal resources [12,13,14,15,16]. Thus, we believe that wearing masks may likely encourage voice behavior.

Drawing on self-perception theory, this study argues that employees’ mask-wearing might positively affect their voice behavior and introduces psychological safety as an underlying mediation mechanism. Self-perception theory indicates that individuals’ internal reflections on their actions in interaction contexts (e.g., wearing a mask) can influence their subsequent behavioral outcomes [17,18]. In the workplace, employees often remain silent because of their uncertainty about whether their suggestions will be approved and endorsed, as well as their fear of negative evaluations and personal costs [6,7,9]. Masks can cover a large part of the face and help conceal wearers’ identities, which promotes the internal reflections on interpersonal situations as less risky and worrying, and consequently reduces the difficulties of engaging in voice behavior. This internal reflection process is closely related to psychological safety, defined as perceptions of showing the authentic self without fear of negative consequences [19].

In summary, this study investigates the relationship between mask-wearing and voice behavior, focusing on the mediating effect of psychological safety (Figure 1). It makes several contributions to the literature. First, by exploring the psychological and behavioral consequences of employees’ mask-wearing in the workplace, we extend the research on the effect of mask-wearing beyond its benefits in people’s health conditions and everyday life. Second, we explore how a seemingly minor behavioral habit like wearing masks in the workplace changes employees’ voice behavior, which can bring about improvement and change to the organization. Third, we draw on self-perception theory and identify psychological safety as a key mechanism explaining why masks can promote wearers’ voice behavior in the workplace.

## 2. Hypothesis Development

Since the outbreak of COVID-19, masks have become a necessity in everyday life. During the pandemic, wearing masks was a public health policy implemented for pandemic prevention, particularly serving as a safety norm within the workplace. After the pandemic, people have continued to wear masks in their everyday lives and work, even in the absence of compulsory regulations. We argue that employees’ voluntary mask-wearing promotes their voice behavior via psychological safety. Psychological safety refers to the extent to which individuals are able to exhibit and express their authentic selves in the workplace without fear of negative consequences [19]. Researchers suggest that psychological safety reflects a sense of safety when individuals engage in self-expression and interpersonal risk-taking, such that employees would not be worried about resentment or rejection when they make mistakes and speak up with different opinions [20].

We draw on self-perception theory [17] to explain the relationship between mask-wearing and psychological safety. According to this theory, individuals ascertain their internal states—emotions, thoughts, and attitudes—through the introspection of their own actions and the context of these actions. Specifically, engaging in an action prompts a self-attribution process, in which individuals reflect on this behavior and consider the conditions that lead them to behave in this way [21]. In line with this reasoning, employees who wear masks in the workplace are likely to feel about their internal states (e.g., perceived safety) by inferring from observations of their mask-wearing and the associated contexts. We argue that mask-wearing behavior can lead to the reflection on internal states of psychological safety in three ways.

First, mask-wearing is recognized as both a societal norm and a workplace safety protocol. During the pandemic, masks have become widely accepted as a preventative measure, and evolved to be viewed as a symbol of collective responsibility and ethical conduct, representing the commitment to communal well-being and sacrificing one’s personal convenience for the sake of public welfare, especially in countries that value collectivism [22]. Existing research has found that masks may reduce wearers’ deviance by increasing their moral awareness [5]. Therefore, when wearing masks, individuals might consider themselves as adhering to a secure social norm, which will enhance their perceptions of psychological safety.

Second, masks may also serve as a self-protective ritual for individuals. Psychological research indicates that rituals, characterized by fixed, repetitive patterns, can evoke emotional resonance and stimulate positive emotions, thereby boosting confidence and self-esteem, while helping individuals alleviate stress and anxiety [23,24]. Therefore, the act of wearing a mask can be considered as a self-protective ritual [25], potentially augmenting individuals’ sense of control and psychological safety.

Third, the relationship between mask-wearing and psychological safety can also be explained through deindividuation. Masks cover a large part of the face and help conceal wearers’ identities [26]. Existing research indicates that wearing masks enhances individuals’ sense of anonymity, which in turn diminishes social visibility and social desirability concerns, contributing to a significant increase in psychological safety [27]. It provokes the perception that the situation is safe for self-expression and interpersonal risk-taking. Therefore, we propose the following hypothesis:

**Hypothesis** **1.**
*Employees’ mask-wearing has a positive effect on their psychological safety.*


Psychological safety can lead to further behavioral consequences in the workplace. Voice behavior, conceptualized as the discretionary expression of constructive ideas, opinions, and concerns about work-related issues, is pivotal for continuous improvement and effective organizational decision-making [6,8,28]. Despite the benefits, employees are often reluctant to engage in voice behavior, as it demands high resource consumption and potential risks such as undermining their credibility and receiving a negative performance evaluation [12,29,30]. Psychological safety refers to the extent to which individuals believe they would not be punished or misunderstood when they engage in risky behaviors, such as speaking up with different and challenging ideas or suggestions [19,31]. When employees are free of fears and concerns about expressing their opinions, the perceived costs and risks of voice would be minimized. In contrast, low psychological safety suppresses open expression and communication of ideas and opinions as employees worry about being criticized or misunderstood. In line with this reasoning, psychological safety has been thought to facilitate voice because such perceptions reduce the felt risk associated with presenting ideas [20,32]. Therefore, we propose the following hypothesis:

**Hypothesis** **2.**
*Psychological safety has a positive effect on voice behavior.*


Building upon Hypotheses 1 and 2, we argue that employees’ mask-wearing will promote their voice behavior via psychological safety. Self-perception theory indicates that individuals interpret their own internal states—such as emotions, motivations, and attitudes—through reflection on their actions and the contexts in which these actions are embedded [17]. Employees who wear masks in the workplace are likely to perceive this behavior as a means of self-protection. This perception not only fosters a sense of security against health risks but also imbues a broader sense of psychological safety, encompassing the freedom to express oneself without fear of negative consequences. When employees feel psychologically safe, they are more inclined to express their opinions, as they trust that their contributions will be received in a supportive manner. Thus, the augmented sense of psychological safety generated from mask-wearing, can further encourage employees to engage in voice behavior in their workplace. This is consistent with empirical research, which has demonstrated that psychological safety plays a mediating role in the dynamics between various contextual factors (such as leadership) and employee voice behavior [31,33]. Therefore, we propose the following hypothesis:

**Hypothesis** **3.**
*Psychological safety mediates the positive relationship between employees’ mask-wearing and voice behavior.*


In sum, this study shifts the focus from how mask-wearing influences everyday interpersonal processes to how mask-wearing influences workplace proactive behavior, primarily through the lens of self-perception theory, considering mask-wearing as a self-protective act which can trigger wearers’ reflection on psychological safety and, in turn, promote voice behavior in the workplace.

The specific objectives of this study are to (a) examine the relationships among mask-wearing, psychological safety and voice behavior, and (b) verify the mediating role of psychological safety in the pathway from mask-wearing to voice behavior. The present study proposes a mediation model (Figure 1). Through this investigation, we seek to fill the existing research gaps and provide nuanced understandings of the effect of wearing masks on individuals’ perceptions and behaviors in organizational contexts.

## 3. Methods

We conducted an online experiment in China to test our hypotheses, using a within-subject design. We chose to focus on China for several reasons. First, mask-wearing is predominantly a public health concern in China, whereas it has been a politically divisive issue in some other countries, which have witnessed anti-mask campaigns [22,34,35,36]. Chinese citizens are used to being asked to put on masks in daily life and at their workplaces. Even after the pandemic, many of them continue to maintain the habit of wearing masks in their daily lives. Second, Chinese people tend to value superficial harmony, which is aligned with a self-protective and defensive orientation [37,38]. In such a culture, individuals are concerned about how others perceive them and their actions, which makes it appropriate to examine the self-concealing effect of masks.

### 3.1. Participants

A total of 300 working adults were recruited from the Credemo platform, a large-scale experiment platform in China. Nine of them were excluded because they failed the attention check, leaving a final sample of 291 participants. Among the 291 participants, 46.0% were male and 54.0% were female, with an average age of 26.1 years (SD = 4.97). The age distribution of participants is as follows: 21 aged less than 20 (7.2%), 122 aged 21–25 (41.9%), 109 aged 26–30 (37.5%), 25 aged 31–35 (8.6%), 10 aged 36–40 (3.4%), and 4 aged older than 40 (1.4%). Participants all had working experience and engaged in face-to-face work communication in their workplaces, with employment in different organizations, including governments (5.2%), public institutions (26.8%), private enterprises (38.8%), state-owned enterprises (24.0%), and foreign-funded enterprises (5.2%).

### 3.2. Procedure and Materials

We conducted an online vignette experiment using a within-subject design. Participants were presented with two distinct scenarios in which they were asked to imagine themselves as members of a sales team during a quarterly review meeting. In this meeting, team members would first summarize their individual work achievements and then give suggestions for improvements.

In Scenario 1, participants were instructed to envision that they are attending this meeting without wearing a mask. Accompanying this instruction, a scenario image depicted the participant in a non-mask-wearing state (as illustrated in Figure 2). Participants were then informed of various dissatisfactions within their team, such as unreasonable job division, poor work atmosphere, and low efficiency from certain colleagues. After reading the scenario, participants completed an attention check. Then they completed a survey that measured their psychological safety and voice behavior.

After this survey, we asked participants to read Scenario 2, which mirrored Scenario 1 with one critical variation: participants were prompted to imagine that they are attending the same meeting, but this time, wearing a mask. In the image, we depicted a natural workplace setting where several team members are wearing masks, rather than only the speaker. The scenario’s accompanying image reflected this masked state (as shown in Figure 3). After reading the scenario, participants also completed an attention check. Then they completed the same survey that measured their psychological safety and voice behavior.

### 3.3. Measures

We used 5-point Likert scales ranging from 1 (“strongly disagree”) to 5 (“strongly agree”) for all substantive variables. We employed translation and back-translation procedures [39] to translate all English items into Chinese.

Psychological safety was measured using the 5-item scale developed by Liang et al. [32] in the Chinese context. Sample items include “In my work unit, I can express my true feelings regarding my job” and “In my work unit, I can freely express my thoughts”. The Cronbach’s α coefficient was 0.82 in Scenario 1 and 0.77 in Scenario 2.

Voice behavior was measured using the 10-item scale developed by Liang et al. [32] in the Chinese context, which includes 5 items of promotive voice and 5 items of prohibitive voice. Sample items include “Proactively develop and make suggestions for issues that may influence the unit” and “Dare to voice out opinions on things that might affect efficiency in the work unit, even if that would embarrass others”. The Cronbach’s α coefficient is 0.92 in Scenario 1 and 0.92 in Scenario 2.

Table 1 presents the scales of psychological safety and voice behavior.

## 4. Results

Table 2 presents the descriptive statistics and correlations of the main variables under investigation.

First, the within-subject effect of mask-wearing on psychological safety and voice behavior was tested with paired-sample *t*-tests. Mediation effects were probed with the MEMORE macro developed by Montoya and Hayes [40].

### 4.1. Within-Subject Main Effects

We used a paired sample *t*-test to analyze the difference in voice behavior under mask-wearing and non-mask-wearing conditions. In support of our prediction, participants reported that they were more likely to engage in voice behavior when they imagined wearing a mask in the conference (M = 3.85, SD = 0.82) than not wearing a mask in the conference (M = 3.74, SD = 0.82), with d = 0.14, *t*(290) = 2.58, and *p* = 0.01. This suggests that employees’ mask-wearing has a positive effect on their voice behavior.

A paired-sample *t*-test was also employed to analyze the difference in psychological safety under mask-wearing and non-mask-wearing conditions. Results are shown in Table 3. As predicted, participants reported that they perceived higher psychological safety when they imagined wearing a mask in the conference (M = 3.56, SD = 0.82) than not wearing a mask in the conference (M = 3.42, SD = 0.87), with d = 0.17, *t*(290) = 3.22, and *p* = 0.001. This suggests that employees’ mask-wearing has a positive effect on their psychological safety.

### 4.2. Mediation Effects

We followed the procedure described by Montoya and Hayes [40] for testing mediation in within-subject designs. Specifically, we used the MEMORE macro to test whether the positive effect of mask-wearing on voice behavior can be mediated by psychological safety. Since the independent variable does not actually exist in the data, the effect of the independent variable would be reflected in the impact of the difference in the mediating variable on the difference in the dependent variable between the two conditions, while controlling for the average of the mediating variable in both conditions. As shown in Table 4, the effect of the difference in psychological safety on the difference in voice behavior between non-mask-wearing and mask-wearing conditions were significant and positive (b = 0.68, SE = 0.04, *p* < 0.001, 95% CI = [0.60, 0.77]), and the difference of voice behavior between non-mask-wearing and mask-wearing conditions was not significant from 0 (b = 0.02, SE = 0.03, *p* > 0.05, 95% CI = [−0.05, 0.08]). This indicated that psychological safety fully mediated the relationship between mask-wearing and voice behavior. In addition, bootstrapping analysis with 5000 interactions showed that the indirect effect of mask-wearing on voice behavior via psychological safety was significant (b = 0.10, SE = 0.03, 95% CI = [0.04, 0.16]). Therefore, our hypotheses were all supported.

## 5. Discussion

Since the pandemic, mask-wearing has influence individual behaviors and social interactions in profound ways. Drawing on self-perception theory, this study examined the positive effect of mask-wearing in the workplace on wearers’ voice behavior through psychological safety. An online experiment using a within-subject manipulation of wearing masks supported our hypotheses.

Firstly, empirical results revealed that the act of wearing a mask can lead to a stronger sense of psychological safety, which supported Hypothesis 1. Consistent with prior studies, masks can help people reduce fears of negative evaluation and social anxiety during interpersonal interactions [2,3,4]. Simultaneously, masks serve as a symbol of moral responsibility, enhancing wearers’ awareness of ethical behavior [5]. The self-protective and moral signaling function of masks results in wearers’ increased psychological safety.

In addition, we found that psychological safety positively affected voice behaviors in the workplace. Echoing the findings of previous studies [20,32], we note that psychological safety lowers the perceived risks associated with voice behavior, encouraging employees to speak up and challenge the status quo. In environments where fears of backlash are minimized, employees are more likely to contribute ideas and feedback.

Furthermore, this study found that psychological safety mediates the relationship between mask-wearing and voice behavior, confirming Hypothesis 3. Although numerous studies have delved into the mediating role of psychological safety in the relationship between contextual factors and employee voice behavior [31,33], it is important to note that some physical actions and practices might also influence employees’ perceptions of psychological safety and further voice behavior. This study, being the first to leverage and extend the self-perception theory [17] within the context of mask-wearing at work, provides a unique perspective in explaining why masks can promote wearers’ voice behavior. Our research suggests that minor behavioral habits such as mask-wearing can significantly shape psychological states and further workplace proactive behaviors.

Our study encourages a broader perspective on mask-wearing in the workplace and has important practical implications for employees and managers. Mask-wearing can be viewed as distant, disrespectful and uncooperative, and can lead to misunderstandings because of inadequate facial expressions and dissatisfying communication experiences [41,42,43]. Contrary to these views, our research reveals the positive aspects of mask-wearing. Our empirical results inspire managers in organizations to view mask-wearing as a positive behavior that enhances employees’ psychological safety and proactive behaviors such as voice. Thus, maintaining an equitable and respectful stance towards employees who choose to wear masks voluntarily could facilitate a more inclusive environment and encourage a richer diversity of employee perspectives.

Nevertheless, this study is not without its limitations. Firstly, our research methods can be enhanced. Despite employing an experimental approach, the online nature and within-subject design call for improvements in internal and external validity. Future research could benefit from mixed methods and varied measures, such as adopting a between-subject design to mitigate sequence effects and measuring actual voice behavior instead of intentions. Secondly, future research should explore additional outcomes of mask-wearing. We only examined the positive effect of mask-wearing on wearers’ voice behavior, it is also worth studying how mask-wearing can influence wearers’ other behaviors with contradictory nature, such as unethical pro-organizational behavior [44]. Meanwhile, in the workplace, how observers (e.g., leaders and coworkers) would perceive and respond to mask-wearing behavior can also be explored. Moreover, the moderating effects of contextual factors such as leadership, the status of mask wearers, as well as individual traits, on the relationship between mask-wearing and voice behavior, should be further investigated. Thirdly, our exploration of boundary conditions was limited. Since this study was conducted in a Chinese context, it is uncertain if our findings are generalizable to other countries and cultural settings. Traditional Chinese values concerning high power distance may make voice a particularly risky behavior [45]. Meanwhile, the self-protective function of masks might be more salient in Chinese culture since mask-wearing has been a public health concern in China rather than some other countries, which have witnessed anti-mask campaigns [22,36]. Furthermore, the Chinese cultural emphasis on maintaining harmonious interpersonal relationships may have increased the relative importance of psychological safety in predicting voice behavior in our sample. We recommend that future research should delve into the cultural contingencies, assessing the universality or specificity of our findings across different cultural contexts.

## 6. Conclusions

Drawing on self-perception theory, this study examined the positive influence of mask wearing in workplace on wearers’ voice behavior through psychological safety. An online experiment using a within-subject manipulation of wearing masks supported our hypotheses.

This study addressed a significant gap in the existing literature by delving into the effects of mask-wearing within workplace environments. While previous studies have explored the psychological impacts of mask-wearing in daily life, little attention has been paid to its effects in the workplace settings. Our empirical results suggest seemingly mundane behaviors such as mask-wearing can have profound effects on employees’ engagement in positive organizational behaviors. In conclusion, this study sheds light on the broader psychological and behavioral implications of mask-wearing, extending beyond its recognized health and mental benefits, and contributing to the understanding of workplace dynamics in the post-pandemic era.

## Figures and Tables

**Figure 1 behavsci-14-00309-f001:**
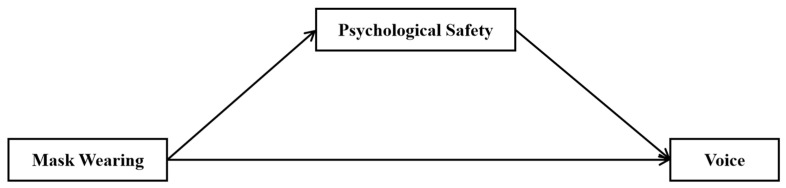
The theoretical model.

**Figure 2 behavsci-14-00309-f002:**
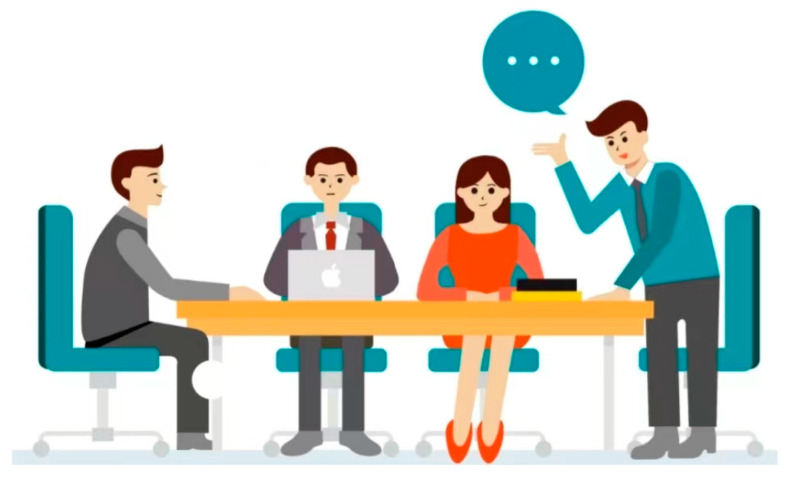
Image in Scenario 1 (non-mask-wearing condition). The picture was designed by the authors.

**Figure 3 behavsci-14-00309-f003:**
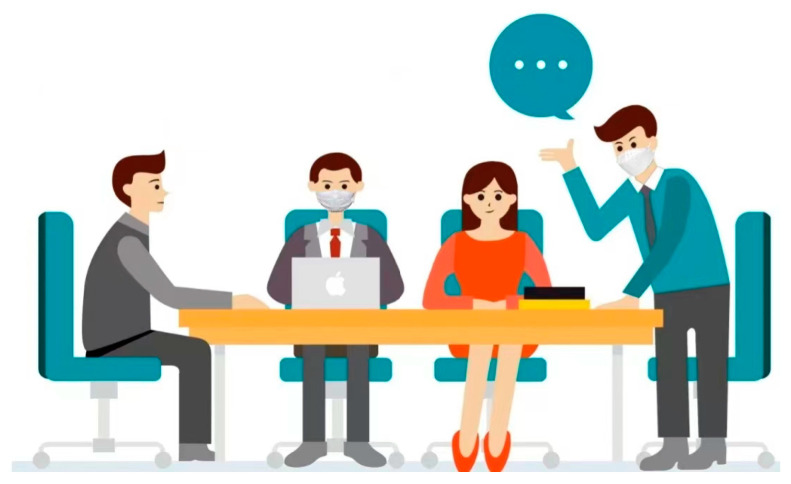
Image in Scenario 2 (mask-wearing version). The picture was designed by the authors.

**Table 1 behavsci-14-00309-t001:** Scales of psychological safety and voice behavior.

Variables	Items
Psychological safety	In my work unit, I can express my true feelings regarding my job.
	In my work unit, I can freely express my thoughts.
	In my work unit, expressing your true feelings is welcomed.
	Nobody in my unit will pick on me even if I have different opinions.
	I’m worried that expressing true thoughts in my workplace would do harm to myself (reverse-coded).
Promotive voice	Proactively develop and make suggestions for issues that may influence the unit.
	Proactively suggest new projects which are beneficial to the work unit.
	Raise suggestions to improve the unit’s working procedure.
	Proactively voice out constructive suggestions that help the unit reach its goals.
	Make constructive suggestions to improve the unit’s operation.
Prohibitive voice	Advise other colleagues against undesirable behaviors that would hamper job performance.
	Speak up honestly with problems that might cause serious loss to the work unit, even when/though dissenting opinions exist.
	Dare to voice out opinions on things that might affect efficiency in the work unit, even if that would embarrass others.
	Dare to point out problems when they appear in the unit, even if that would hamper relationships with other colleagues.
	Proactively report coordination problems in the workplace to the management.

**Table 2 behavsci-14-00309-t002:** Mean, standard deviation, and correlations of research variables.

Variables	M	SD	1	2	3	4	5	6
1. Gender	0.46	0.499	1					
2. Age	26.08	4.966	0.144 *	1				
3. Psychological safety (Scenario 1)	3.417	0.873	0.153 **	0.029	1			
4. Psychological safety (Scenario 2)	3.559	0.820	0.161 **	0.039	0.610 **	1		
5. Voice (Scenario 1)	3.738	0.821	0.272 **	−0.003	0.739 **	0.405 **	1	
6. Voice (Scenario 2)	3.850	0.821	0.240 **	0.020	0.526 **	0.741 **	0.593 **	1

Note. N = 291. * *p* < 0.05, ** *p* < 0.01.

**Table 3 behavsci-14-00309-t003:** Paired-sample *t*-test results.

	Mask-Wearing M (SD)	Non-Mask-Wearing M (SD)	d	*t*	df	*p*
Voice behavior	3.85 (0.82)	3.74 (0.82)	0.14	2.58	290	0.01
Psychological safety	3.56 (0.82)	3.42 (0.87)	0.17	3.22	290	0.001

Note. N = 291.

**Table 4 behavsci-14-00309-t004:** Mediation effect test results.

	b	SE	95% CI
Mask-wearing → Psychological safety	0.68	0.04	[0.60, 0.77]
Mask-wearing → Voice behavior (with mediator)	0.02	0.03	[−0.05, 0.08]
Indirect effect	0.10	0.03	[0.04, 0.16]

Note. N = 291.

## Data Availability

All data included in the current study can be obtained from the corresponding author upon reasonable request.

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
