# Peer review of "Veiled to Express: Uncovering the Effect of Mask-Wearing on Voice Behavior in the Workplace"

_behavsci, 2024, doi:10.3390/bs14040309_

Round 1
Reviewer 1 Report
Comments and Suggestions for Authors
Thank you for providing me with the opportunity to review this manuscript.
The manuscript is well written and explores an interesting topic.
Nevertheless, I would like to recommend to the authors improve the discussion section. It would be beneficial for them to delve into a more comprehensive analysis of their findings in relation to existing literature. Specifically, the authors should explicitly articulate whether their findings align with previous studies. Furthermore, it would be advantageous for them to reiterate key arguments to elucidate the interpretation of the results obtained.
Reviewer 2 Report
Comments and Suggestions for Authors
Dear authors,
You approached a particularly interesting subject, an original approach. The article captivates the reader and brings new information to the scientific community.
However, I believe that the article needs improvement in order to reach the level required by the academic community.
Line 19 - State the type of consequences, otherwise it may be implied that the mask has negative psychological and behavioral effects.
Line 23 - 33 - The introduction is far too short. I recommend the introduction of several paragraphs dealing with the impact of COVID, the behavioral changes of people, wearing the mask, disinfecting, isolation, etc., so that the wearing of the mask should be introduced gradually, along with the other psychological effects.
Line 37 - Replace the word speculate with a synonym, because it induces a state in which assumptions are made, not scientific measurements.
Line 58 - Words are repeated, please rephrase.
Line 119 - Replace "we now turn" with a more objective expression eg. it turns
Line 146 - You must also explain hypothesis 3.
Line 147 - At the end of section 2, I recommend the introduction of a graphic abstract, which summarizes the information and hypotheses. At the same time, I believe that the purpose, objectives, research questions and working hypotheses should be clearly repeated.
Line 163 - The average is not relevant for the age segment. Two 26-year-old people are totally different from an 18-year-old and a 34-year-old. Write exactly the subjectivity number by age interval.
Line 190 - Cite correctly the place where you got the figure. Are there no copyright issues?
Line 199 - 208 - If there are so few items, 5 and 10, I recommend putting them in a table and appearing in the text.
Line 210 - In the results, the counted answers should actually appear.
Line 238 - Data tables can be entered. I recommend that the data be interpreted more, be exploited more.
Line 247 - At Discussions, the statistical interpretation appears, and the analysis is based on the 3 hypotheses, each of which will be discussed and tested.
Line 247 - The results based on the bibliography must also be discussed during the discussions.
Line 253 - 273 - I think that the conclusion works.
Line 313 - Conclusions are almost non-existent. I recommend moving the previously mentioned paragraph.
Good luck!
Round 2
Reviewer 2 Report
Comments and Suggestions for Authors
Congratulations!